# Acceptability of Dry Dog Food Visual Characteristics by Consumer Segments Based on Overall Liking: a Case Study in Poland

**DOI:** 10.3390/ani8060079

**Published:** 2018-05-23

**Authors:** David Gomez Baquero, Kadri Koppel, Delores Chambers, Karolina Hołda, Robert Głogowski, Edgar Chambers

**Affiliations:** 1Center for Sensory Analysis and Consumer Behavior, Kansas State University, 1310 Research Park Drive, Manhattan, KS 66502, USA; dagomezb@ksu.edu (D.G.B.); delores@ksu.edu (D.C.); eciv@ksu.edu (E.C.); 2Department of Animal Breeding and Production, Warsaw University of Life Sciences, Ciszewskiego 8, Warsaw 02-786, Poland; karolina_holda@sggw.pl (K.H.); robert_glogowski@sggw.pl (R.G.)

**Keywords:** affective tests, consumer acceptance, consumer perception, appearance, dry dog food, kibble, cluster analysis

## Abstract

**Simple Summary:**

The present work aimed to explore the response of dog owners to the appearance of pet foods. The objective of this study was to understand the impact of the visual characteristics of dry dog food on the human consumers’ acceptance and beliefs they associate with the products. The acceptability of the appearance of dry dog foods by consumers is influenced by the number of different kibbles present, color(s), shape(s), and size(s) in the product. The results indicated that dry dog food manufacturers should take special consideration with the appearance of the kibbles to enhance the acceptability of their products. These findings can help dry dog food manufacturers meet the consumers’ needs with increasing benefits to the pet food and commodity industries.

**Abstract:**

Sensory analysis of pet foods has been emerging as an important field of study for the pet food industry over the last few decades. Few studies have been conducted on understanding the pet owners’ perception of pet foods. The objective of this study is to gain a deeper understanding on the perception of the visual characteristics of dry dog foods by dog owners in different consumer segments. A total of 120 consumers evaluated the appearance of 30 dry dog food samples with varying visual characteristics. The consumers rated the acceptance of the samples and associated each one with a list of positive and negative beliefs. Cluster Analysis, ANOVA and Correspondence Analysis were used to analyze the consumer responses. The acceptability of the appearance of dry dog foods was affected by the number of different kibbles present, color(s), shape(s), and size(s) of the kibbles in the product. Three consumer clusters were identified. Consumers rated highest single-kibble samples of medium sizes, traditional shapes, and brown colors. Participants disliked extra-small or extra-large kibble sizes, shapes with high-dimensional contrast, and kibbles of light brown color. These findings can help dry dog food manufacturers to meet consumers’ needs with increasing benefits to the pet food and commodity industries.

## 1. Introduction

The pet food industry represents an important sector of the food processing industry. In 2015, global pet food retail sales reached 70 billion USD with 20 billion USD corresponding to the combined market in Europe [1]. By 2021, the dry pet food market in Europe is anticipated to reach over 11 billion USD [2]. In 2017, the total pet care sales for Eastern Europe are estimated to reach 5.43 billion USD. Poland represents the second largest pet food market in Eastern Europe accounting for 14.82% of the regional market share [3]. Poland’s dog ownership is among the highest in the world. Approximately, 45% of Polish population lives with dogs [4] for an estimated total of 7.8 million dogs living in Polish households [5]. The pet food market in Poland increased 9% in 2014 and represented an estimated total of 639 million USD in 2015 [5]. Poland was ranked the eighth highest dry dog food producing country in the world in 2015 with 584,000 metric tons [1].

In the pet food industry, the development of successful products depends on a wide variety of factors. As with human foods, the development of new products in this sector must consider both the nutritional and the sensory aspects of the product. From the sensory perspective, the use of sensory analysis methods is key to gain understanding of pet and owner behavior and to provide manufacturers and researchers the means to study pet food selection. Nevertheless, research publications regarding the sensory characteristics of pet food products are relatively new, as much of the work conducted previously seems to be proprietary. As described by Koppel [6], the use of sensory evaluation methods to study pet foods can be accomplished using humans, animals and instruments. Most of the published work on sensory analysis of pet foods using humans has focused on using descriptive sensory analysis methods to study the characteristics of the products. Di Donfrancesco et al. [7] developed a lexicon to describe the appearance, aroma, flavor and texture characteristics of dry dog food products using a trained human sensory panel. Koppel et al. [8] studied the effect of fiber inclusion on the sensory characteristics and palatability of dry dog food products. In addition to dry dog food, Pickering conducted studies to describe the flavor and texture characteristics of dry and wet cat foods using a human sensory panel [9,10]. Some studies have been published on studying the pet owners’ response to pet food products. Tengpongsathon and Phaosathienpan [11] studied the importance of brand, price, type of food and nutrition on the consumers’ preferences for pet foods in Thailand. Several works have been published on studying the attitudes of pet owners towards pet food. Boya et al. [12] studied how the choice of dog food varies across different dog owners’ segments and the similarity between the dog owner’s criteria at the time of purchasing food for themselves vs. the criteria used when purchasing dog food. Michel et al. [13] investigated feeding practices and attitudes dog and cat owners have towards pet foods and diets they use for feeding their companion animals. Tesfom and Birch [14] studied similarities in the way dog owners buy food for their dogs vs. food for themselves.

As pet owners, humans make decisions at the time of purchasing pet food products. It is common for dog food manufacturers to try to develop foods that satisfy the owners’ requirements as much as they do with the pet requirements. From a sensory point of view, the product’s success depends on the companion animal accepting the product as palatable. In addition, the pet owner’s perception of the product is of great importance since the owner makes the purchase decision. From a sensory perspective, the interaction dog owners have with dog food is usually through the senses of sight and smell. Di Donfrancesco et al. [15] studied the overall acceptability, aroma acceptability and appearance acceptability of dry dog food products by consumers in the United States. The results showed that the appearance is more important than the aroma in driving the consumers’ liking of dry dog food products. A wide variety of colors, geometric shapes, sizes and kibble mixtures can be found in the dry dog food market. Companies strive to catch the customers’ attention by developing products with innovative visual characteristics to make their products stand out over the competition. Given the results by Di Donfrancesco et al. [15], further study is necessary to gain a deeper understanding on what kind of visual characteristics are preferred by consumers and what are some of the factors driving appearance liking by consumers in dry fog food products. Koppel et al. [16] studied this subject for consumers in Thailand and found that Thai consumers liked kibbles with a bone shape and yellowish color best.

To address the study of consumer response to the appearance of dry dog food products, the present work aimed: (1) to understand the impact of visual characteristics of dry dog food on human consumers’ acceptance and beliefs; (2) to identify potential differences in the preferred visual attributes by consumer segments in Poland based on their acceptance of the appearance of dry dog foods; (3) to gain a deeper understanding of the impact of different visual characteristics on the overall acceptability by the consumers; and (4) to study the association between the visual characteristics and the beliefs consumers link to dry dog foods.

## 2. Materials and Methods

### 2.1. Samples

Dry dog food samples (n = 30) of kibbles from commercially available dry dog foods were used. The samples were prepared by selecting specific kibbles from a wide list of commercial products to use them as a single-kibble sample or by mixing different kibbles to create multiple-kibble samples. The samples were chosen to represent a wide variety of visual characteristics in terms of colors, sizes, shapes and number of kibbles present in the samples (Table 1). To classify the size of the kibbles, a relative size scoring method was used ranging from a relative size score of 1 (extra-small size) to 7 (extra-large size). Kibbles with similar colors were grouped together into general color categories to facilitate the analysis of the data. The shape of the samples is described as an approximation to common 3D shapes. All the commercial products were purchased in local pet stores/grocery stores in the Manhattan, Kansas, US, prior to the test and after selection and preparation were stored under frozen conditions until the day of testing. All the products were evaluated within the “best by” date and no recalled products were used.

### 2.2. Participants

The participants were screened to be: (1) 18 years of age or above; (2) dog owners; (3) to use dry dog food to feed their dog(s); (4) to be responsible for purchasing the dog food or to participate in making the purchase decision on which food is fed to the dog(s); and (5) not to have been diagnosed with color vision deficiencies previously. A total of 120 participants were recruited and participated voluntarily in the study. The demographics of the participants are shown in Table 2.

### 2.3. Consumer Study

Consumer testing was performed in compliance with the Kansas State University (KSU) Institutional Review Board #7710. A Central Location Test (CLT) was conducted at Warsaw University of Life Sciences (Warsaw, Poland). Participants were recruited from the metropolitan area of Warsaw via e-mail, phone, social media, flyers, and word-of-mouth. Test sessions were conducted using a classroom setting and lasted 45 min. The number of participants at each session ranged from 1 to 14. The consumers did not receive an incentive for participating in the study.

The samples were presented monadically to the consumers using a randomized Latin square design [17]. Samples were presented in white 8-oz cups Styrofoam^®®^ containers covered with lids and labeled with three-digit codes.

### 2.4. Questionnaires

Each consumer was presented with one demographic questionnaire and 30 sets of dog food questionnaires. The participants completed the demographic questionnaire prior to sample evaluation. Next, consumers were asked to visually inspect each of the samples presented and to answer one set of the dog food questionnaires for each of the products. The participants were asked to rate the Overall Liking, Size Liking, Shape Liking and Color Liking for each of the samples using a 9-point hedonic scale (ranging from 1 = “dislike extremely” to 9 = “like extremely”, 5 = “neither like nor dislike”). After the hedonic questions, participants were presented with a list of thirteen positive and negative functional terms in a check-all-that-apply question and asked to select all those they associated with each of the samples. The following 5 positive and 8 negative terms were used:Positive terms: “Has natural ingredients/raw materials”, “Good for dog’s health”, “My dog will like it”, “Has variety of ingredients/raw materials”, and “Has all the nutrients that my dog(s) needs”.Negative terms: “Looks like fake food”, “Color is too pale”, “Consumption may cause choking hazard”, “My dog will not eat it”, “I don’t like the shape of this sample”, “Has artificial color(s)”, “Has too much variety of shapes”, and “Has too much variety of colors”.

The terms were selected based on previous work conducted and expertise on the topic.

### 2.5. Data Analysis

#### 2.5.1. Cluster Analysis

To group consumers with similar liking patterns given the set of samples, cluster analysis was performed using Agglomerative Hierarchical Clustering (AHC) method and Ward’s agglomeration method on the Overall Liking scores. Demographics were calculated for each resulting cluster.

#### 2.5.2. Analysis of Variance

Two-way Analysis of Variance (ANOVA) was performed to model each of the four acceptance attributes Overall Liking, Size Liking, Shape Liking and Color Liking (dependent variable) as a function of Sample and Consumer (explanatory variables) using a 95% level of significance. Tukey’s Honest Significant Difference (HSD) pairwise comparison tests were performed for the ANOVA models using Sample as factor for pairwise comparisons to determine significant differences among samples for each acceptance attribute. The Analysis of Variance models for each of the four hedonic attributes were performed: (1) across all 120 consumers; and (2) for each of the consumer clusters.

#### 2.5.3. Correspondence Analysis

To analyze the results from the check-all-that-apply (CATA) question, a contingency table was constructed by summarizing the times a term was checked by the consumers for each of the thirty samples and for each of the Overall Liking clusters. Chi-square distance was used to test the independence between samples and terms using a level of significance α = 0.05. Correspondence Analysis was used to study the association between samples and attributes and to display the results in two-dimensional maps.

All statistical analyses were performed using XLSTAT Version 2015.3.01 (Addinsoft, New York, NY, USA).

## 3. Results

### 3.1. ANOVA of Overall Liking Scores

There was evidence of a significant effect by the two explanatory variables (sample and consumer) on the Overall Liking, Size Liking, Shape Liking, and Color Liking mean scores for all the participants and for each of the consumer clusters (Table 3). The results from the Type III SS indicate a significant effect of the sample on the average score for Overall Liking, Size Liking, Shape Liking, and Color Liking in all cases (Table 4).

#### 3.1.1. Most Liked Samples Overall

A high degree of discrimination was found among the sample set by the consumers (Table A1a). The mean scores presented a range of 2.6 in the hedonic scale (minimum mean score = 3.7; maximum mean score = 6.3). Samples rated highest for Overall Liking included: (1) single-kibble samples with colors in the shades of brown color category (golden brown, medium brown), medium kibble sizes, low-dimensional contrast kibbles and traditional kibble shapes such as triangular prisms (S14, mean score = 6.3; S15, mean score = 6.3), cuboids (S4, mean score = 5.6) and flat cylinders (S13, mean score = 5.4); (2) single-kibble samples with colors in the shades of brown color category (bright gold, golden brown, medium brown), large kibble sizes, low-dimensional contrast kibbles and innovative kibble shapes such as the bones (S1, mean score = 5.4), the cylindrical ‘X’ (S3, mean score = 5.6) and the rack of ribs (S16, mean score = 5.9). According to the Tukey’s HSD test, all these scores were found not to be significantly different to each other at the 95% confidence level.

Consumers overall rated lowest for Overall Liking: (1) single-kibble samples with colors in the shades of brown color category (amber brown, light brown), kibble sizes ranging from medium-to-large to extra-large, and a high-dimensional contrast kibble shape such as the sticks (S5, mean score = 3.7) and the discs (S6, mean score = 3.8); (2) a single-kibble sample of extra-dark brown color, a low-dimensional contrast kibble shape (spheres) and an extra-small kibble size (S18, mean score = 3.8); (3) a single-kibble sample of medium brown color with holes present in the center of the kibbles (S12, mean score = 4.1); (4) single-kibble samples of green colors (S19, mean score = 4.0; S20, mean score = 3.9); (5) a single-kibble sample of red color (S21, mean score = 3.8); and (6) multiple-kibble samples containing kibbles with some of the previous characteristics - the high-dimensional contrast discs and kibbles with holes present in the center (M1, mean score = 4.2; M3, mean score = 3.8). According to the post-hoc test, all these scores were found not to be significantly different to each other at the 95% confidence level (Table A1a).

#### 3.1.2. Size Liking

A high degree of discrimination was found among the consumers for size liking (Table A1a). The average scores presented a range of 3.3 in the hedonic scale (minimum mean score = 3.1; maximum mean score = 6.4). Samples rated highest include single-kibble samples of: (1) medium size (S4, mean score = 5.8; S15, mean score = 6.0); (2) medium-to-large size (S14, mean score = 6.4; S17, mean score = 5.8); and (3) large size (S2, mean score = 5.7; S3, mean score = 6.2; S16, mean score = 6.0).

The consumers overall rated lowest for Size Liking single-kibble samples of: (1) extra-small size (S18, mean score = 3.1); (2) small size (S8, mean score = 3.3); (3) small-to-medium size (S11, mean score = 4.3; S12, mean score = 4.0; S13, mean score = 4.3); and (4) extra-large size (S5, mean score = 3.3; S7, mean score = 4.3). Significant differences were found among these scores according to the post-hoc test, with the score of sample S18 being not significantly different than the scores of samples S5 and S8 only.

#### 3.1.3. Shape Liking

A moderate degree of discrimination was found among the sample set (Table A1a). The mean scores presented a range of 3.3 in the hedonic scale (minimum mean score = 3.2; maximum mean score = 6.5). Samples rated highest overall for Shape Liking included: (1) single-kibble samples of low-dimensional contrast kibbles and traditional shapes such as triangular prisms (S14, mean score = 6.5; S15, mean score = 6.3), cuboids (S4, mean score = 6.2; S17, mean score = 5.9), flat cylinders (S13, mean score = 5.8) and puffs (S19, mean score = 5.7); (2) single-kibble samples with an innovative kibble shape such as the cylindrical ‘X’ (S3, mean score = 5.8) and the rack of ribs (S16, mean score = 5.8); and (3) multiple-kibble samples containing kibbles with low-dimensional contrast and traditional kibble shapes such as cuboids (M2, mean score = 5.9) and a mixture of cuboids, flat cylinders and puffs (M6, mean score = 5.6).

Samples rated lowest for Shape Liking overall included: (1) single-kibble samples with a high-dimensional contrast kibble shape such as the sticks (S5, mean score = 3.2) and the discs (S6, mean score = 3.8); (2) single-kibble samples with holes present in the center of the kibbles (S9, mean score = 4.1; S12, mean score = 4.1); and (3) multiple-kibble samples containing kibbles with high-dimensional contrast (discs) and holes present in the center (M1, mean score = 3.8; M3, mean score = 3.9).

#### 3.1.4. Color Liking

For Color Liking, a high degree of discrimination was found among the set of samples (Table A1a). The average scores presented a range of 3.2 in the hedonic scale (minimum mean score = 3.4; maximum mean score = 6.6). Samples rated highest by consumers overall include single-kibble samples of medium brown colors (S11, mean score = 6.0; S13, mean score = 5.6; S14, mean score = 6.2; S15, mean score = 6.6; S16, mean score = 5.9). According to the Tukey’s HSD test, only the score of sample S13 was found to be significantly lower to the score of sample S15 at the 95% confidence level.

Samples rated lowest for Color Liking by consumers overall included: (1) a single-kibble sample of red color (S21, mean score = 3.4); (2) single-kibble samples of green colors (S19, mean score = 3.5; S20, mean score = 3.5); (3) a single-kibble sample of light brown color (S6, mean score = 3.6); (4) multiple-kibble samples with a high-color contrast containing kibbles of red, green and shades of brown colors (M6, mean score = 3.8; M8, mean score = 3.9); and (5) a multiple-kibble sample containing kibbles with two of the previous characteristics—light brown and green colors (M3, mean score = 3.7).

### 3.2. Analysis by Consumer Clusters

#### 3.2.1. AHC Analysis

Three consumer clusters were obtained with the following distribution of participants as shown in Table A1b,c,d. Cluster 1 had the smallest number of consumers with only 12.5% of the participants (15), cluster 2 represented 33.3% of the consumers (40) and cluster 3 included the highest number of assessors with 54.2% (65). The demographics of the consumer clusters are shown in Table 5.

#### 3.2.2. Cluster 1

Although significant differences were found for the Overall Liking, a low degree of discrimination was found among the samples in cluster 1 (Table A1b). Sample S1 showed the highest average score (mean score = 7.6), despite being not significantly different from following samples S11 (mean score = 7.5) and S14, S16 and S2 (mean score = 7.3). Sample S21 had a significantly lower score (mean score = 5.0) than sample S1, but not significantly different from the other 28 samples. All scores were above the neutral category (neither like nor dislike = 5.0), which shows a high level of acceptability by consumers in cluster 1 for all 30 samples. Samples rated highest in Overall Liking are single-kibble samples with colors in the shades of brown category (from bright gold to medium brown), medium sizes (from small-to-medium to large), and with either traditional shapes (triangular prisms) or more innovative shapes (bones, clovers, rack of ribs, flat ‘X’).

Consumers in cluster 1 overall rated lowest for Overall Liking single-kibble samples with a distinctive visual characteristic from the pool of samples, such as: (1) red color (S21, mean score = 5.0); (2) extra-dark brown color and extra-small size (S18, mean score = 5.5); and (3) a high-dimensional contrast kibble shape such as the sticks (S5, mean score = 5.6) and the discs (S6, mean score = 5.6).

In terms of size, a higher degree of discrimination was found when compared to the other three acceptance attributes. Samples S1 and S16 showed the highest score (mean score = 7.2) and were rated significantly higher than samples S18 (mean score = 4.7) and S7 (mean score = 4.4). Samples rated highest for Size Liking include: (1) single-kibble samples with kibble sizes in the medium-to-large range such as medium (S15, mean score = 7.1), medium-to-large (S1, mean score = 7.2; S14, mean score = 7.1), and large (S16, mean score = 7.2; S22, mean score = 7.1); and (2) a multiple-kibble sample containing kibbles with sizes ranging from medium to large (M7, mean score = 7.0).

Consumers in cluster 1 rated lowest for Size Liking single-kibble samples with sizes in the two ends of the size scale such as extra-small size (S18, mean score = 4.7), small size (S8, mean score = 4.9) and extra-large size (S7, mean score = 4.4; S5, mean score = 5.1).

A low degree of discrimination was found among the samples. Sample S1 showed the highest score (mean score = 8.0) and was rated significantly higher than sample S5 (mean score = 5.3) only. Samples rated highest for Shape Liking include single-kibble samples with innovative shapes such as bones (S1, mean score = 8.0), clovers (S2, mean score = 7.6), cylindrical ‘X’ (S3, mean score = 7.5), steaks (S22, mean score = 7.4), and flat cuboids with center hole (S12, mean score = 7.4). Most of the samples rated highest are in the shades of brown color category, except for the red meat and white fat steaks (S22).

Samples rated lowest for Shape Liking by consumers in cluster 1 included: (1) single-kibble samples with a high-dimensional contrast such as the sticks (S5, mean score = 5.3) and the discs (S6, mean score = 6.0); and (2) single-kibble samples with a low-dimensional contrast such as the cylinders (S7, mean score = 5.7) and the spheres (S18, mean score = 5.7).

For Color Liking, a low degree of discrimination between the samples was found among the samples. Sample S1 showed the highest average score (mean score = 7.4) and was significantly higher than sample S21 (mean score = 4.6) only. Samples rated highest in Color Liking are single-kibble samples with shades of brown colors such as bright gold (S1, mean score = 7.4), medium brown (S14, mean score = 7.2; S15, mean score = 7.1; S11, mean score = 7.1), and golden brown (S8, mean score = 7.1).

Samples rated lowest for Color Liking in cluster 1 include: (1) single-kibble samples of red (S21, mean score = 4.6), light brown (S6, mean score = 5.1), dark green (S20, mean score = 5.5), and extra-dark brown (S18, mean score = 5.5) colors; and (2) a multiple-kibble sample with a high-color contrast containing shades of brown, green and red colors (M6, mean score = 5.3).

#### 3.2.3. Cluster 2

A high degree of discrimination was found among the samples by consumers in cluster 2 (Table A1c). Samples rated highest for Overall Liking include: (1) single-kibble samples with innovative shapes such as the rack of ribs (S16, mean score = 6.4) and the steaks (S22, mean score = 6.4); (2) single-kibble samples with traditional shapes such as flat triangular prisms (S15, mean score = 6.0); and (3) multiple-kibble samples with high-color contrast and low-dimensional contrast such as M6 (mean score = 6.3), M2 (mean score = 6.1) and M8 (mean score = 6.0).

Consumers in cluster 2 rated lowest for Overall Liking: (1) single-kibble samples in the shades of brown color category with a distinctive visual characteristic from the pool of samples such as extra-dark color and extra-small size (S18, mean score = 2.7), a hole present in the middle of the kibble (S9, mean score = 3.4; S12, mean score = 3.7), and a high-dimensional contrast kibble shape such as the discs (S6, mean score = 3.8); and (2) a multiple-kibble sample containing two of the previously mentioned characteristics (M1, mean score = 3.9).

In terms of size, a high degree of discrimination was found. Samples rated highest for Size Liking include single-kibble samples of: (1) medium-to-large kibble size (S14, mean score = 6.4); and (2) large sized kibbles (S3, mean score = 6.4; S16, mean score = 6.3; S22, mean score = 6.3).

Samples rated lowest for Size Liking by consumers in cluster 2 include single-kibble samples of: (1) extra-small size (S18, mean score = 2.7); (2) small size (S8, mean score = 2.8); (3) small-to-medium size (S11, mean score = 3.6; S13, mean score = 3.7; S12, mean score = 3.8); and (4) extra-large size (S5, mean score = 3.4).

A high degree of discrimination was found among the sample set. Samples rated highest in terms of Shape Liking include: (1) single-kibble samples with innovative shapes such as the rack of ribs (S16, mean score = 6.5), the cylindrical ‘X’ (S3, mean score = 6.2), and the steaks (S22, mean score = 6.2); (2) single-kibble samples with more traditional-looking shapes such as rounded triangular prisms (S14, mean score = 6.2); and (3) multiple-kibble samples containing low-dimensional contrast kibbles such as a mixture of rounded cuboids (M2, mean score = 6.2), and a mixture of rounded cuboids, puffs, rounded triangular prisms, and flat cylinders (M8, mean score = 6.2).

Samples rated lowest for Shape Liking in cluster 2 include: (1) the extra-small spheres (S18, mean score = 3.6); (2) single-kibble samples with high-dimensional contrast such as the sticks (S5, mean score = 3.7) and the discs (S6, mean score = 4.0); (3) single-kibble samples with holes present such as the flat cuboids with center hole (S12, mean score = 3.7) and the flat triangular prisms with center hole (S9, mean score = 4.0); and (4) a multiple-kibble sample with high-dimensional contrast containing discs and flat triangular prisms with center hole (M1, mean score = 3.9).

For Color Liking, a high degree of discrimination among the samples was found. Samples rated highest include: (1) single-kibble samples with medium brown colors such as S15 (mean score = 6.1), S16 (mean score = 6.0), S11 (mean score = 5.8) and S14 (mean score = 5.8); (2) the innovative red meat and white fat raw steak-like kibbles (S22, mean score = 5.9); and (3) multiple-kibble samples with high-color contrast such as combination of golden brown and red (M2, mean score = 5.8), and combination of golden brown, dark brown and red (M4, mean score = 5.8).

Samples rated lowest for Color Liking by consumers in cluster 2 include: (1) single-kibble samples in the shades-of-brown color category such as light brown (S6, mean score = 3.4), extra-dark brown (S18, mean score = 3.5), medium brown (S12, mean score = 3.5; S9, mean score = 3.7), and dark brown (S17, mean score = 3.9); (2) the single-kibble dark green sample (S20, mean score = 3.8); and (3) a multiple-kibble sample in the shades of brown color category with low-color contrast (M1, mean score = 3.7).

#### 3.2.4. Cluster 3

A high degree of discrimination was found from consumers in cluster 3 (Table A1d). Samples rated highest for Overall Liking included single-kibble samples with colors in the shades of brown category (from golden brown to medium brown), medium sizes (from small-to-medium to medium-to-large) and traditional-looking shapes such as triangular prisms (S14, mean score = 6.3; S15, mean score = 6.3), flat cylinders (S13, mean score = 5.5), and rounded cuboids (S4, mean score = 5.4).

Samples rated lowest for Overall Liking in cluster 3 include: (1) single-kibble samples with high-dimensional contrast such as the sticks (S5, mean score = 2.9); (2) single-kibble samples of red color (S21, mean score = 2.8) and green color (S19, mean score = 3.1; S20, mean score = 3.1); and (3) multiple-kibble samples with high-color-contrast containing green (M3, mean score = 2.9; M5, mean score = 3.2), red (M4, mean score = 3.0), and green and red colors (M6, mean score = 3.1; M8, mean score = 3.2).

For Size Liking, a high degree of discrimination was found from the consumers. Samples rated highest include single-kibble samples of: (1) medium kibbles sizes (S15, mean score = 5.9; S4, mean score = 5.7); (2) medium-to-large size (S14, mean score = 6.3; S17, mean score = 5.8); and (3) large size (S3, mean score = 5.9).

Samples rated lowest for Size Liking in cluster 3 include single-kibble samples of: (1) extra-small size (S18, mean score = 3.1); (2) small size (S8, mean score = 3.3); (3) small-to-medium size (S12, mean score = 3.6); and (4) extra-large size (S5, mean score = 2.9; S7, mean score = 3.7).

A high degree of discrimination was found among the samples. Samples rated highest include single-kibble samples with traditional-looking shapes such as triangular prisms (S14, mean score = 6.6; S15, mean score = 6.2), rounded cuboids (S4, mean score = 6.3), cuboids (S17, mean score = 6.0), and flat cylinders (S13, mean score = 5.9).

Samples rated lowest for Shape Liking in cluster 3 include: (1) single-kibble samples with high-dimensional contrast such as the sticks (S5, mean score = 2.4) and the discs (S6, mean score = 3.1); and (2) multiple-kibble samples with different characteristics fall in this category, all of which contain discs and flat triangular prisms with center hole (M3, mean score = 3.0; M7, mean score = 3.1; M1, mean score = 3.2; M5, mean score = 3.2).

For Color Liking, a high degree of discrimination was found among the sample set. Samples rated highest include single-kibble samples with medium brown color (S15, mean score = 6.9; S14, mean score = 6.3; S13, mean score = 5.9; S11, mean score = 5.8).

Consumers in cluster 3 rated lowest for Color Liking: (1) single-kibble samples of red (S21, mean score = 2.3; S22, mean score = 2.7) and green (S19, mean score = 2.7) color; and (2) multiple-kibble samples with high-color contrast where the red and/or green colors are present (M6, mean score = 2.4; M8, mean score = 2.4; M4, mean score = 2.5).

#### 3.2.5. Correspondence Analysis

A difference on the distribution of the functional terms consumers linked to the samples on each cluster was found (Table 6).

##### Cluster 1

Total of 63.12% of the total variation is explained by the first two dimensions (Figure 1a). Positive terms such as “Has all the nutrients that my dog(s) needs”, “Good for dog’s health”, “My dog will like it” and “Has natural ingredients/raw materials” were found close to each other on the correspondence analysis (CA) map and were associated with: (1) single-kibble samples (S1, S4, S9, S11, S14, S15, S16, and S17) to which colors in the shades of brown category and medium sizes are characteristic; and (2) multiple-kibble samples with low-color contrast (M1 and M5).

Samples S5 (extra-large size) and S18 (extra-small size) were associated with the term “Consumption may cause choking hazard”. Samples S21, M2 and M4 contained red kibbles and were related with the term “Looks like fake food”. Samples S21, S22, M2, M4, M6, and M8 were associated with the term “Has artificial color(s)”, and they all had red kibbles present. As expected, the high-color contrast multiple-kibble sample M6 was related with the term “Has too much variety of colors”. Samples S2 (bright gold), S3 (golden brown), S6 (light brown), S7 (from bright gold to medium brown), and S8 (golden brown) were associated with the term “Color is too pale”. Samples S3 (cylindrical ‘X’), S6 (discs), S7 (cylinders), and S8 (cuboids) were related with the term “I don’t like the shape of this sample”.

##### Cluster 2

Total of 61.34% of variation was explained by the first two dimensions (Figure 1b). All 5 positive beliefs were associated by consumers in cluster 2 with: (1) single-kibble samples (S14, S15, S16 and S17) to which medium-to-dark brown colors, medium-to-large sizes and traditional shapes (except for S16) are characteristic; and (2) a high-color-contrast and low-dimensional-contrast multiple-kibble sample (M2).

Samples S6 (light brown) and M1 (light brown, medium brown) were associated with the term “Color is too pale”. As expected, samples M5 (discs, flat triangular prisms, flat triangular prisms with center hole, puffs) and M7 (clovers, discs, cylindrical ‘X’, flat triangular prisms, flat triangular prisms with center hole) were related with the term “Has too much variety of shapes”. Samples S7 (cylinders), S8 (cuboids), S9 (flat triangular prisms with center hole), S11 (flat ‘X’), S12 (flat cuboids with center hole) and S13 (flat cylinders) are associated with the terms “I don’t like the shape of this sample” and “My dog will not eat it”. Samples S18 (extra-small size) and S5 (extra-large size) were related with the term “Consumption may cause choking hazard”. As expected, high-color-contrast multiple-kibble samples (M4, M6 and M8) were associated with the term “Has too much variety of colors”. Samples containing red kibbles (S21, S22, M2, M4, M6 and M8) and green kibbles (S19, S20, M6 and M8) were related with the term “Has artificial color(s)”. Samples containing green (S19 and S20), red (S22) and amber brown (S10) were associated with the term “Looks like fake food”.

##### Cluster 3

Total of 76.47% of variation was explained by the first two dimensions (Figure 1c). All 5 positive beliefs were found close to each other in the CA map and were associated with single-kibble samples such as S13, S14, S15 and S17 to which medium-to-dark brown colors, medium sizes, traditional shapes (cuboids, triangular prisms, flat cylinders) and low-dimensional contrast kibbles are characteristic. Two more innovative-looking samples such as the medium brown and large size rack of ribs (S16) and the extra-small size and extra-dark brown spheres (S18) were also related with all five positive terms.

The terms “Has artificial color(s)” and “Looks like fake food” were associated with: (1) single-kibble samples of red (S21 and S22), green (S19 and S20) and amber brown (S10) colors; and (2) multiple-kibble samples with high-color-contrast containing red (M2, M4, M6, and M8) and green (M6 and M8) colors. As expected, high-color-contrast multiple-kibble samples (M2, M4, M6 and M8) were related with the term “Has too much variety of colors”. Samples S6 (light brown) and M1 (light brown, medium brown) were associated with the term “Color is too pale”. Samples S8 (small size), S3 (large size) and S7 (extra-large size) are related with the term “Consumption may cause choking hazard”. The term “I don’t like the shape of this sample” was associated with: (1) samples containing kibbles with disc shapes (S6 and M1); and (2) single-kibble samples with innovative shapes such as holes in the center (S9, S12 and M1), bones (S1), cylindrical ‘X’ (S3) and cylinders (S7). As expected, the three multiple-kibble samples with the highest variety in terms of shapes (M3, M5 and M7) were related with the term “Has too much variety of shapes”. The term “My dog will not eat it” was associated with: (1) single-kibble samples with innovative shapes such as clovers (S2) and sticks (S5); and (2) multiple-kibble samples with a high variety of shapes (M3, M5 and M7).

## 4. Discussion

The results of this research showed that the acceptability of the appearance of dry dog food by consumers is affected by the number of kibbles, color(s), shape(s), and size(s) present in the product. These results complement the results by Di Donfrancesco et al. [15] who found that the color of the kibbles and the size of them affect the liking of dry dog food by consumers. The participants overall showed preference for single-kibble samples of brown colors, medium kibble sizes, and traditional kibble shapes such as triangular prisms. It should be noted that samples liked best overall presented a low-dimensional contrast kibble shape which is in agreement with the results by Di Donfrancesco et al. [15] who found that samples containing kibbles with a high uniformity of shape were liked better than samples with kibbles with a low uniformity of shape. The consumers overall disliked kibbles of extra-small size and with the darkest brown color in the sample set which is in accordance with the results by Di Donfrancesco et al. [15] and Koppel et al. [16]. In addition, kibbles of extra-large size were disliked by the participants overall which was reported by Koppel et al. [16]. Also, kibbles of light brown color were disliked overall by the participants which was previously found by Di Donfrancesco et al. [15]. In addition, the participants overall did not rate multiple-kibble samples containing variety of colors, shapes and sizes highest. In contrast, previous studies found that multiple-kibble samples were well received by the consumers in the US [15] and in Thailand [16] which shows an interesting difference in the liking of the visual characteristics of dry dog food by consumers in different countries. Furthermore, Koppel et al. [16] found that consumers in Thailand liked best a single-kibble sample with a bone shape and received well dry dog food with non-traditional kibble shapes which differs from the findings of the present study. However, some similarities in the preferences of consumers towards the appearance of dry dog food are identified across countries which provide guidelines pet food manufacturers could use as a basis for product development targeting different markets on a global scale. Previous research has shown the similarities/differences in the consumer perception and preferences when testing consumer products across different countries [18,19,20,21]. This shows the importance of understanding the needs and preferences of the target consumers on each market to achieve the development of successful products. The results of this research and the differences found across countries in the liking of the appearance of dry dog food evidences the importance of conducting further research on specific markets to accomplish a successful marketing of pet foods.

As explained by Koppel [6], there are a number of factors that affect the purchase decision of dry dog food which include price, brand, packaging, advertising claims, nutritional value and ingredients, and specific characteristics of the product such as appearance (number of different kibbles, color(s), shape(s), size(s)) and aroma. Also, the dog’s response to the product and the amount consumed by the companion animal plays a key role on influencing the purchase decision, along with the health benefits perceived by the owner and the digestive effect and characteristics of the stool. For this reason, repurchase of a dry dog food product depends on its ability to meet the pet owners’ and the companion animal’s needs along all the previously mentioned factors. Specific visual characteristics of dry dog food that are perceived as satisfactory by the target population can increase the overall consumers’ degree of satisfaction with the product and improve the chance of repurchase. However, manufacturers should strive to meet the consumers’ requirements for all the factors influencing the purchase decision of dry dog food.

The results of the Correspondence Analysis showed that consumers associate specific visual characteristics with positive beliefs such as “Has natural ingredients”, “Good for dog’s health”, “Has variety of ingredients” and “Has all the nutrients that my dog(s) needs”. Likewise, the consumers related specific visual characteristics with negative beliefs. Kumcu and Woolverton [22] found that premium human food purchasers are more likely to purchase premium pet food for their pets. This raises a question of whether dry dog food with visual characteristics that are well received by the consumers and that is associated with positive beliefs is seen as being more premium quality by the consumers. Furthermore, the same question can be made for pet food other than dry dog food as the pet food market continues to diversify following the trend of humanization of pet foods. This could be an interesting topic to address in further research projects. In addition, Kumcu and Woolverton [22] found that young consumers are more likely to purchase premium pet foods despite budget constraints. This tendency may be expected to persist and perhaps even to grow in the future as young consumers age. As the pet food market continues to grow following the humanization trend, consumers are demanding more specialized premium pet foods.

Samples containing red kibbles were perceived as “Looks like fake food” and “Has artificial color(s). Genschow et al. [23] concluded in a previous study with human foods that red color functions as a subtle stop signal that works even outside of a person’s focused awareness and may thereby reduce incidental intake of foods and drinks. In another study, Bruno et al. [24] found that red plates reduced the consumption of food and the use of hand cream, while the liking towards the samples presented similar scores from all the plates. In addition, Bruno et al. also proved that their results were neither dependent on the Michelson (luminance) contrast nor on the color contrast either, which led them to suggest that the effect of the red color on consumption might simply be due to avoidance associated with the color of the plate that was influencing the participants [24]. This human behavior that has been reported previously regarding red color may not just be limited to human foods, but it could also extend to the perception of humans towards pet foods. This could be an interesting topic for further research on human perception of the visual characteristics of pet foods.

The present study took place in Warsaw, Poland, and Polish citizens from the metropolitan area of Warsaw participated in the sessions. It should be pointed out that the results of this research represent a good estimation for urban consumers in Poland but do not necessarily represent the preferences for consumers in rural areas of Poland. However, the analysis performed using consumer clusters showed the results from different consumer groups and enables a better representation of the variation that can be found in the Polish market. In addition, Poland represents the second largest pet food market in Eastern Europe [3] and the findings from this study may be of good use by manufacturers who market their products in Eastern Europe as a basis for product development.

## 5. Conclusions

The degree of liking of the appearance of dry dog food samples by dog owners is influenced by the number of different kibbles present, color(s), shape(s), and size(s) in the product. The results indicate that dry dog food manufacturers should take special consideration with the visual characteristics of the kibbles to satisfy the pet owners’ expectations and to enhance the acceptability of their products. It is recommended for dry dog food manufacturers who market their products in Poland to prioritize the production of single-kibble samples of brown colors (from golden brown to medium brown), medium kibble sizes, and traditionally-looking kibble shapes such as triangular prisms. Likewise, it is recommended for dry dog food companies to avoid the production of kibbles with a high-dimensional-contrast shape such as discs and sticks, extra-small or extra-large sized kibbles, and the use of light brown colors.

## Figures and Tables

**Figure 1 animals-08-00079-f001:**
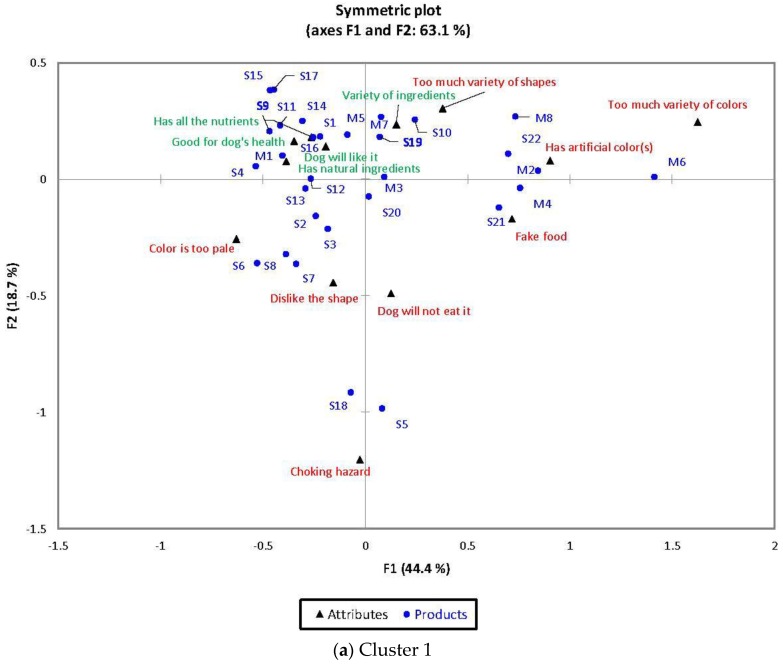
Correspondence Analysis maps between all thirty samples and all thirteen functional terms from the check-all-that-apply question. Positive terms shown in green; negative terms shown in red.

**Table 1 animals-08-00079-t001:** Description of the samples (n = 30) used and their visual characteristics.

Sample	Sample Type	Number of Kibbles Present	Color(s)	Relative Size(s) Score (1–7)	Shape(s)
S1	Single-kibble	1	Bright gold	5	Bones
S2	Single-kibble	1	Bright gold	6	Clovers
S3	Single-kibble	1	Golden brown	6	Cylindrical ‘X’
S4	Single-kibble	1	Golden brown	4	Rounded cuboids
S5	Single-kibble	1	Amber brown	7	Sticks
S6	Single-kibble	1	Light brown	5	Discs
S7	Single-kibble	1	Shades of brown (from bright gold to medium brown)	7	Cylinders
S8	Single-kibble	1	Golden brown	2	Cuboids
S9	Single-kibble	1	Medium brown	4	Flat triangular prisms with center hole
S10	Single-kibble	1	Amber brown	4	Puffs (irregular)
S11	Single-kibble	1	Medium brown	3	Flat ‘X’
S12	Single-kibble	1	Medium brown	3	Flat cuboids with center hole
S13	Single-kibble	1	Medium brown	3	Flat cylinders
S14	Single-kibble	1	Medium brown	5	Rounded triangular prisms
S15	Single-kibble	1	Medium brown	4	Flat triangular prisms
S16	Single-kibble	1	Medium brown	6	Rack of ribs
S17	Single-kibble	1	Dark brown	5	Semi-flat cuboids
S18	Single-kibble	1	Extra-dark brown	1	Spheres
S19	Single-kibble	1	Medium green	3	Puffs
S20	Single-kibble	1	Dark green	6	Flat elongated cuboids with rounded corners
S21	Single-kibble	1	Red	3	Rounded cuboids
S22	Single-kibble	1	Red meat and white fat, marbled	6	Steaks
M1	Multiple-kibble	2	Light brown, Medium brown	4, 5	Discs, flat triangular prisms with center hole
M2	Multiple-kibble	2	Golden brown, Red	3, 4	Rounded cuboids
M3	Multiple-kibble	3	Light brown, Medium green, Medium brown	3, 4, 5	Discs, flat triangular prisms with center hole, puffs
M4	Multiple-kibble	3	Golden brown, Dark brown, Red	2, 3, 4	Rounded cuboids, flat cylinders
M5	Multiple-kibble	4	Light brown, Medium green, Medium brown	3, 4, 5	Discs, flat triangular prisms with center hole, puffs, flat triangular prisms
M6	Multiple-kibble	4	Golden brown, Dark brown, Medium green, Red	2, 3, 4	Rounded cuboids, flat cylinders, puffs
M7	Multiple-kibble	5	Golden brown, Bright gold, Light brown, Medium brown	4, 5, 6	Clovers, discs, cylindrical ‘X’, flat triangular prisms, flat triangular prisms with center hole
M8	Multiple-kibble	5	Bright gold, Golden brown, Dark brown, Medium green, Red	2, 3, 4, 5	Rounded cuboids, flat cylinders, puffs, rounded triangular prisms

A relative size scoring method ranging from 1 (smallest size) to 7 (largest size) was used to classify the size of the kibbles as follows: 1 = extra-small; 2 = small; 3 = small-to-medium; 4 = medium; 5 = medium-to-large; 6 = large; 7 = extra-large. All relative sizes were assigned based on the largest dimension for each of the kibbles, regardless of the shape.

**Table 2 animals-08-00079-t002:** Summary of the demographics of the participants in the consumer study (%).

**Gender**	**Male**					**Female**	
	40.8%					59.2%	
**Age (years)**	**18–34**					**35 or above**	
	41.7%					58.3%	
**Number of dogs owned**	**1**		**2**		**3**	**4**	**5 or more**
	64.2%		22.5%		6.7%	3.3%	3.3%
**Size of dog(s) (can choose more than one answer if more than one dog)**	**Very small (0.5–5 kg)**		**Small (5.1–11.0 kg)**		**Medium (11.1–20.0 kg)**	**Large (20.1–40.0 kg)**	**Very large (more than 40 kg)**
	5.8%		33.3%		36.7%	39.2%	9.2%
**Money spent on each dog per month (USD)**	**Less than $15**		**$15–$50**			**More than $50**	
	13.3%		68.3%			18.3%	
**Knowledgeable about pet food and pet’s health**	**Yes**					**No**	
	54.2%					45.8%	
**Important factor(s) considered when choosing dog food (can choose up to three answers)**	**Improve dog’s health in general**	**Brand**	**Price**	**Appearance of the product**	**Dog(s) like(s) that food**	**Ingredients/Raw materials**	**Dog(s) need(s) that food because of a health condition**
	30.8%	30.0%	50.8%	10.0%	70.8%	75.0%	6.7%
**Purchasing location (check-all-that-apply)**	**Online**		**Clinic/Veterinary hospitals**	**Small market in living area**	**Supermarkets/Convenience stores**	**Pet shops/Pet stores**	**Market fairs**
	54.2%		15.0%	10.0%	30.0%	44.2%	5.0%

**Table 3 animals-08-00079-t003:** Summary of the two-way ANOVA tests for all the participants and for each of the three consumer clusters. Overall Liking, Size Liking, Shape Liking and Color Liking as dependent variables. Sample and Consumer as explanatory variables. A level of significance α = 0.05 was used.

Parameter	Consumers	Dependent Variable			
		Overall Liking	Size Liking	Shape Liking	Color Liking
**R²**	**All participants**	0.314	0.351	0.367	0.324
	**Cluster 1**	0.214	0.235	0.306	0.300
	**Cluster 2**	0.270	0.315	0.272	0.240
	**Cluster 3**	0.313	0.359	0.400	0.415
**F**	**All participants**	10.652	12.606	13.512	11.179
	**Cluster 1**	2.578	2.907	4.156	4.048
	**Cluster 2**	6.157	7.650	6.212	5.264
	**Cluster 3**	9.079	11.186	13.285	14.184
***p*-value**	**All participants**	<0.0001	<0.0001	<0.0001	<0.0001
	**Cluster 1**	<0.0001	<0.0001	<0.0001	<0.0001
	**Cluster 2**	<0.0001	<0.0001	<0.0001	<0.0001
	**Cluster 3**	<0.0001	<0.0001	<0.0001	<0.0001

**Table 4 animals-08-00079-t004:** Type III Sum of Squares for the two-way ANOVA tests for all the participants and for each of the three consumer clusters. Analysis of the impact of Sample on the model. Overall Liking, Size Liking, Shape Liking and Color Liking as dependent variables. Sample and Consumer as explanatory variables. A level of significance α = 0.05 was used.

Dependent Variable	All Participants		Cluster 1		Cluster 2		Cluster 3	
	F	*p*-Value	F	*p*-Value	F	*p*-Value	F	*p*-Value
**Overall Liking**	18.758	<0.0001	1.9	0.004	11.39	<0.0001	21.24	<0.0001
**Size Liking**	28.371	<0.0001	2.67	<0.0001	12.99	<0.0001	17.06	<0.0001
**Shape Liking**	26.151	<0.0001	2.03	0.002	9.143	<0.0001	23.87	<0.0001
**Color Liking**	20.559	<0.0001	1.66	0.019	7.473	<0.0001	32.57	<0.0001

**Table 5 animals-08-00079-t005:** Summary of demographics from the overall liking clusters (percentage of consumers).

**Gender**	**Cluster Number**	**Male**	**Female**
	Cluster 1	20.0%	80.0%
	Cluster 2	42.5%	57.5%
	Cluster 3	44.6%	55.4%
**Age (years)**	**Cluster Number**	**18–34**	**35 or Above**
	Cluster 1	73.3%	26.7%
	Cluster 2	45.0%	55.0%
	Cluster 3	32.3%	67.7%

**Table 6 animals-08-00079-t006:** Chi-square distance tests of association between samples and terms from the CATA question for all three clusters. A level of significance α = 0.05 was used.

Cluster No.	1	2	3
**Chi-square (Observed value)**	621.092	1327.974	3176.161
**Chi-square (Critical value)**	392.501	392.501	392.501
**DF**	348	348	348
***p*-value**	<0.0001	<0.0001	<0.0001

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
