# Peer review of "Acceptability of Dry Dog Food Visual Characteristics by Consumer Segments Based on Overall Liking: a Case Study in Poland"

_animals, 2018, doi:10.3390/ani8060079_

Round 1

Reviewer 1 Report

The study was very carefully conducted and the results support the conclusions. It was a very interesting paper to read! However, the results are extensive and their presentation is very long. The length and organization of the results section makes the paper challenging to read. It is recommended that:

+ Table 6 be made an appendix

+ Figures 1, 2, and 3 be moved to an Appendix

+A simple Table be added that summarizes the AHC analyses and the Text be reduced significantly to cover the key points shown in the table and refer the reader to more detail in the Appendix

+ the level of detail in the description of the results of the Correspondence analyses also needs reduction

+ The discussion and conclusions read very well .

Author Response

The study was very carefully conducted and the results support the conclusions. It was a very interesting paper to read! However, the results are extensive and their presentation is very long. The length and organization of the results section makes the paper challenging to read. It is recommended that:

+ Table 6 be made an appendix

A: Agree. Table 6 was moved to Appendix A, page 18.

+ Figures 1, 2, and 3 be moved to an Appendix

A: Disagree. The Correspondence Analysis maps are key to illustrate the results from the Correspondence Analysis on this research. The authors recommend showing these maps in the results part of the manuscript where they are being used rather than in an Appendix. Instead of being labeled as Figures 1, 2 and 3, the maps have been re-labeled as Figure 1a, 1b, and 1c to comprise the results of all three consumer clusters in just one figure and to save some space from the captions of Figures 2 and 3.

+A simple Table be added that summarizes the AHC analyses and the Text be reduced significantly to cover the key points shown in the table and refer the reader to more detail in the Appendix

A: Disagree. The purpose of the analysis by consumer clusters (AHC analysis) is to provide in depth information about the specific characteristics that the participants on each of the consumer clusters like and dislike. There are 4 specific acceptance attributes to study for each of the consumer clusters and, given the variety of visual characteristics being compared in terms of number of kibbles, size(s), shape(s), and color(s) present among the sample set, the authors consider it is not feasible to summarize all that information in a simple table without losing a significant amount of valuable information and affecting negatively the quality of the AHC analysis.

+ the level of detail in the description of the results of the Correspondence analyses also needs reduction

A: Disagree. The present study is comprehensive when it comes to the number of visual characteristics being compared and naturally it yields a considerably high amount of data to be analyzed. The authors consider the results from the Correspondence Analysis are displayed in a summarized way for each of the three consumer clusters. The level of detail used is necessary to enable the comparison of the results from the different consumer segments and to extract similarities/differences to be used in the discussion part.

+ The discussion and conclusions read very well

Reviewer 2 Report

Specific comments:

Page 2, line 94  ... understanding of the impact ...

Page 3, line 99  How many pet food companies were represented?

Page 4, Table 1  For "relative sizes score (1-7)", explain the scale in a footnote to this table and then eliminate Table 2

Page 6, line 127  ... participants at each session ...

Page 7, lines 170-171  Are all of the statistical methods mentioned in lines 152-159 found in this XLSTAT reference?  If not, provide a separate reference for the 4 methods.

Page 10, Table 6  Is there any other way of denoting statistical differences among treatments than use of superscript lettering (where it is almost impossible to decipher the differences among treatments)?

Page 14, line 281  ... despite being not significantly ...

Page 21, line 539  It is known that pets much prefer blue and green colors and that they dislike red colors.  Seems like your results agree with this.

Page 21, lines 547 and 552  Polish

Page 22 and 23, References  Be consistent in use of upper and lower case in the manuscript titles.  Some have lower case lettering after the first word, whereas other have all upper case lettering.

General comments:

This is a very well written paper reporting data of high technical merit.  Results will be of interest to many in the pet food industry.  Authors have studied the topic thoroughly so the results obtained are quite robust.  Other than what I've written above, my only other comment is that if authors can figure out a way to summarize the large tables presented in the text and put these particular tables in a "Supplemental Data" section (if one exists for this journal), this would be preferred.  These large tables are almost in the "raw data" category.  I'm not sure how to do this and, indeed, it may be impossible, but authors should give it some thought.

Author Response

Page 2, line 94  ... understanding of the impact ...

A: Agree. Modified on page 2, line 94

Page 3, line 99  How many pet food companies were represented?

A: Disagree. The samples of kibbles were selected from commercially available products, but they were not showed in their original presentation. Kibbles from different commercial products were selected to be used as either single-kibble samples or to be mixed to create multiple-kibble samples but neither original products nor brands were tested in this study. The participants were not given any information about brands or pet food companies used in this study for any of the samples. The purpose of this research was not to test different brands, pet food manufacturers or readily available products.

Page 4, Table 1  For "relative sizes score (1-7)", explain the scale in a footnote to this table and then eliminate Table 2

A: Agree. Table 2 was deleted. The relative size scoring method is explained in the footnote of Table 1, pages 4 and 5.

Page 6, line 127  ... participants at each session ...

A: Agree. Modified on page 6, new line 130.

Page 7, lines 170-171  Are all of the statistical methods mentioned in lines 152-159 found in this XLSTAT reference?  If not, provide a separate reference for the 4 methods.

A: Yes, all the statistical methods were performed using the same XLSTAT software. No modification necessary.

Page 10, Table 6  Is there any other way of denoting statistical differences among treatments than use of superscript lettering (where it is almost impossible to decipher the differences among treatments)?

A: The common way of showing results of pairwise comparisons tests is to add letters to discriminate significant differences among the samples. In the caption of Table 6, it is explained that “for each of the dependent variables, samples not sharing the same letter differ significantly (in each column)”. The samples are organized in descending order of mean overall liking score on each table to make it easier for the reader to interpret the results. The authors do not suggest further changes for showing the results in Table 6 (new Table A1, page 18).

Page 14, line 281  ... despite being not significantly ...

A: Agree. Modified on page 14, new line 271.

Page 21, line 539  It is known that pets much prefer blue and green colors and that they dislike red colors.  Seems like your results agree with this.

A: The results indicate most of the participants in Poland disliked red kibbles. For further research, it would be interesting to compare this result with the preferences of the pets regarding the color and to study similarities/differences between the pets and the pet owners’ acceptance to the color of the pet food.

Page 21, lines 547 and 552  Polish

A: Agree. Modified on page 21, new lines 534 and 539.

Page 22 and 23, References  Be consistent in use of upper and lower case in the manuscript titles.  Some have lower case lettering after the first word, whereas other have all upper case lettering.

A: Agree. Modified on pages 21 and 22.

General comments:

This is a very well written paper reporting data of high technical merit.  Results will be of interest to many in the pet food industry.  Authors have studied the topic thoroughly so the results obtained are quite robust.  Other than what I've written above, my only other comment is that if authors can figure out a way to summarize the large tables presented in the text and put these particular tables in a "Supplemental Data" section (if one exists for this journal), this would be preferred.  These large tables are almost in the "raw data" category.  I'm not sure how to do this and, indeed, it may be impossible, but authors should give it some thought.

A: Agree. Based on feedback from Reviewers 1 and 2, the large tables (former Table 6) have been moved to Appendix A, page 18.